# Genetic control of thermomorphogenesis in tomato inflorescences

Shuai Sun[1,4], Zhiqiang Liu[1,4], Xiaotian Wang[1], Jia Song[1], Siyu Fang[1], Jisheng Kong[1], Ren Li[1], Huanzhong Wang [2] & Xia Cui [3] ✉

Understanding how plants alter their development and architecture in response to ambient temperature is crucial for breeding resilient crops. Here, we identify the quantitative trait locus *qMULTIPLE INFLORESCENCE BRANCH 2* (*qMIB2*), which modulates inflorescence branching in response to high ambient temperature in tomato (*Solanum lycopersicum*). The non-functional *mib2* allele may have been selected in large-fruited varieties to ensure larger and more uniform fruits under varying temperatures. *MIB2* gene encodes a homolog of the *Arabidopsis thaliana* transcription factor SPATULA; its expression is induced in meristems at high temperature. MIB2 directly binds to the promoter of its downstream gene *CONSTANS-Like1* (*SlCOL1*) by recognizing the conserved G-box motif to activate *SlCOL1* expression in reproductive meristems. Overexpressing *SlCOL1* rescue the reduced inflorescence branching of *mib2*, suggesting how the MIB2−SlCOL1 module helps tomato inflorescences adapt to high temperature. Our findings reveal the molecular mechanism underlying inflorescence thermomorphogenesis and provide a target for breeding climate-resilient crops.

Plants respond to temperature by altering their growth rates and meristem identity; this phenotypic plasticity helps their acclimation to environmental conditions[1,2]. Optimizing reproductive meristem plasticity may enhance crop adaptation and production at elevated temperatures. Thermomorphogenesis is the morphological and architectural changes induced by high temperature (HT) and recent studies have identified key regulators of thermomorphogenesis during vegetative development[3–5]. However, mechanistic insights into thermomorphogenesis during reproductive development (such as the effects of HT on signaling to specify floral identity and inflorescence architecture) remain limited.

Fruit weight and inflorescence architecture are major traits that determine yield in tomato (*Solanum lycopersicum*)[6,7]. Understanding temperature sensing and the pathways that contribute to the developmental plasticity of tomato plants is necessary for the rapid selection of resilient tomatoes[1]. Tomato varieties exhibit an exceptionally diverse architecture of inflorescence[8,9]. The regulators, influencing sympodial meristem activity and floral fate during inflorescence development, have been demonstrated to be able to implicate in determining inflorescence branching[7,10–13]. Combination of these genes in different heterozygous and homozygous configurations, it could provide a possibility for improved fruit yield to create hybrid lines with rational fine-tuning of branching[6]. However, for achieving desirable production, favorable inflorescence branching that balances the sink-source relationship of fruit number and size is important for modern tomato varieties[14]. The regulators influencing floral fate and meristem activity during reproductive development have been proved to determine fruit size and inflorescence branching[10,11,13,15,16], but the responses of tomato to changes in ambient temperature are poorly understood.

[1]State Key Laboratory of Vegetable Biobreeding, Sino-Dutch Joint Laboratory of Horticultural Genomics, Institute of Vegetables and Flowers, Chinese Academy of Agricultural Sciences, Beijing 100081, China. [2]Department of Plant Science and Landscape Architecture, University of Connecticut, Storrs, CT 06269, USA. [3]Key Laboratory of Quality and Safety Control for Subtropical Fruit and Vegetable, Ministry of Agriculture and Rural Affairs, College of Horticulture Science, Zhejiang A&F University, Hangzhou, Zhejiang 311300, China. [4]These authors contributed equally: Shuai Sun, Zhiqiang Liu. ✉e-mail: cuixia@caas.cn

In this work, we identify a quantitative trait locus, *qMULTIPLE INFLORESCENCE BRANCH 2* (*qMIB2*), which maintains stable fruit size by modulating inflorescence branching in response to high ambient temperature. The *mib2* allele may have been selected in tomato breeding that results in stability of inflorescence branching at high ambient temperature to ensure larger and more uniform fruits of modern tomato variates. *MIB2* encodes a homolog of the Arabidopsis thaliana transcription factor SPATULA, which is induced by high temperature and binds directly to the promoters of downstream genes *SlCOL1* to promoting their expression. Overexpression of *SlCOL1* rescues the decrease of inflorescence branches in *mib2* mutants. Our findings reveal the molecular mechanisms underlying thermomorphogenesis signaling in inflorescence meristems, and provide an effective candidate for designing climate-resilient crops facing high temperature extremes in the future.

## Results

### *qMIB2* is a regulator of tomato reproductive development

To investigate how tomato fruit size and inflorescence architecture respond to temperature changes, we evaluated fruit weight and inflorescence branching in the tomato cultivar Moneymaker (MM) and the wild cherry tomato accession LA1310 (*S. lycopersicum* var. *cerasiforme*, CC), which generally grows in cooler regions, under different conditions in the field. The weight of CC fruits decreased and inflorescence branching markedly increased in the summer growth cycle (HT conditions in the field; daily mean temperature: 32.2 °C, peak temperature: 38.7 °C) compared to the spring growth cycle (normal temperature [NT] conditions in the field; daily mean temperature: 23.2 °C, peak temperature: 26.2 °C) (Fig. 1a–c, Supplementary Fig. 1a). By contrast, we observed no significant changes in fruit weight or slight increase in inflorescence branching in MM between these conditions. Consistent with the change in inflorescence branching, fruit number significantly increased in CC but not in MM under HT relative to NT (Fig. 1d). Therefore, the effects of HT on fruit and inflorescence development differ between MM and CC.

We evaluated fruit weight and inflorescence branching under NT and HT conditions in a subset of recombinant inbred lines (RILs) derived from a cross between CC and MM[17]. Like the parental lines, the RILs that produced more inflorescence branches under HT conditions had smaller fruits, whereas other RILs with stable inflorescence architecture exhibited no clear changes in fruit weight between the two conditions. We observed a negative correlation between the change in fruit weight and inflorescence branching in these RILs (Supplementary Fig. 1b).

To clone HT-responsive genes, we selected RIL ST082, exhibiting more inflorescence branching and lighter fruits under HT conditions, and crossed it to MM to generate a mapping population (Fig. 1e, f). We identified the quantitative trait locus (QTL) *qMIB2* (*QTL of MULTIPLE INFLORESCENCE BRANCH 2*) on chromosome 2 by bulked-segregant analysis sequencing (BSA-seq) using pools of BC₃F₂ plants with extremely high or low inflorescence branching (Fig. 1g, Supplementary Fig. 1c). We generated two near-isogenic lines (NILs), NIL-*MIB2*^ST082 and NIL-*mib2*^MM, by three continuous backcrossing ST082 into MM to validate the function of this locus. Although NIL-*MIB2*^ST082 plants had more inflorescence branches than NIL-*mib2*^MM plants under both NT and HT conditions, HT increased inflorescence branching more strongly in NIL-*MIB2*^ST082 than in NIL-*mib2*^MM (Fig. 1h, Supplementary Fig. 1d), indicating that *qMIB2* regulates inflorescence branching in response to HT.

Along with the change in inflorescence branching, the average fruit weight of NIL-*MIB2*^ST082 decreased by more than 20% under HT compared to NT conditions, whereas that of NIL-*mib2*^MM only decreased by less than 8%, although there is also significantly change (Fig. 1i). To exclude the possibility that *qMIB2* directly regulates fruit weight, we removed all but five fruits per truss in NIL-*MIB2*^ST082 and NIL-

*mib2*^MM plants, finding no significant difference in fruit weight between these NILs (Supplementary Fig. 1e), suggesting that *qMIB2* does not directly affect fruit development. Thus, *qMIB2* regulates inflorescence plasticity in response to HT.

### *qMIB2* is selected during tomato breeding

Since inflorescence branching and fruit weight are important agronomic traits that determine tomato yield, we investigated whether the *qMIB2* locus was selected during tomato domestication. We analyzed the genomic sequences of 290 natural tomato accessions, comprising 142 large-fruited accessions (fruit weight = 111.3 ± 68.2 g), 99 cherry accessions (fruit weight = 13.3 ± 9.5 g), and 49 *S. pimpinellifolium* accessions (fruit weight = 2.0 ± 0.85 g)[18], we determined that *qmib2*^MM is only present in some of these accessions (Fig. 1j). Compared to accessions with the functional *qMIB2*^ST082 locus, the average fruit weight was larger in accessions with the non-functional *qmib2*^MM locus (Fig. 1k). We selected three accessions with *qmib2*^ST082 (TS-111, TS-160, TS-9) or *qMIB2*^MM (TS-137, TS-43, TS-51) to examine whether inflorescence branching and fruit number or weight were altered in response to HT. Inflorescence branching and fruit number were less sensitive to HT in the *qmib2*^MM accessions. These accessions produced fewer branches and fruits under HT conditions compared to the accessions with *qMIB2*^ST082 (Supplementary Fig. 2a, b). In line with the changes in inflorescence branching and fruit number, fruit weight was more stable in *qmib2*^MM than in *qMIB2*^ST082 accessions under different temperature conditions (Supplementary Fig. 2c). Therefore, the *qmib2*^MM locus might be selected during tomato breeding to reduce inflorescence branching and maintain fruit size under field conditions.

### Map-based cloning of *MIB2*

We used 960 individuals of the BC₃F₂ population to clone *qMIB2*, which we ultimately narrowed down to a 55.92-kb region containing eight annotated genes (Fig. 2a). Nucleotide polymorphism revealed *Solyc02g093280* as a candidate gene for this locus (Fig. 2a). An insertion in the second exon of *Solyc02g093280* leads to early translation termination in MM. The protein encoded by *Solyc02g093280* shares high sequence similarity with the Arabidopsis (*Arabidopsis thaliana*) basic helix-loop-helix (bHLH) transcription factor SPATULA (SPT) (Supplementary Fig. 3a), which promotes carpel fusion and vegetative organ expansion[19–22]. Aligning well with its roles in inflorescence branching, *MIB2* was extensively expressed in various meristem tissues, especially floral meristems (Supplementary Fig. 3b, c).

To verify the function of MIB2 in inflorescence development, we transformed MM with a complementation construct containing the coding region of *Solyc02g093280* driven by its native promoter (*proMIB2:MIB2-YFP-HA*). The independent complementation lines *proMIB2:MIB2-YFP-HA* L1 and *proMIB2:MIB2-YFP-HA* L2 displayed stronger responses to HT than the MM controls. Specifically, the number of inflorescence branches was markedly higher in these complementation plants under HT than NT conditions (Fig. 2b, c), suggesting that MIB2 positively regulates HT-induced inflorescence branching. Along with increased inflorescence branching, the number of fruits per truss significantly increased two-fold in *proMIB2:MIB2-YFP-HA* plants under HT conditions (Fig. 2d), whereas fruit weight significantly decreased under HT (Fig. 2e), compared with the transgenic line at NT. By contrast, inflorescence branching was reduced in the knockout lines *mib2cr-1* and *mib2cr-2*, which were generated using clustered regularly interspaced short palindromic repeat (CRISPR)/CRISPR-associated nuclease 9 (Cas9)-mediated editing in the NIL-*MIB2*^ST082 background. The increased inflorescence branching induced by HT was suppressed in the *mib2cr* mutants (Fig. 2f, g and Supplementary Fig. 3d, e). These results demonstrate that MIB2 positively regulates inflorescence branching in response to HT.

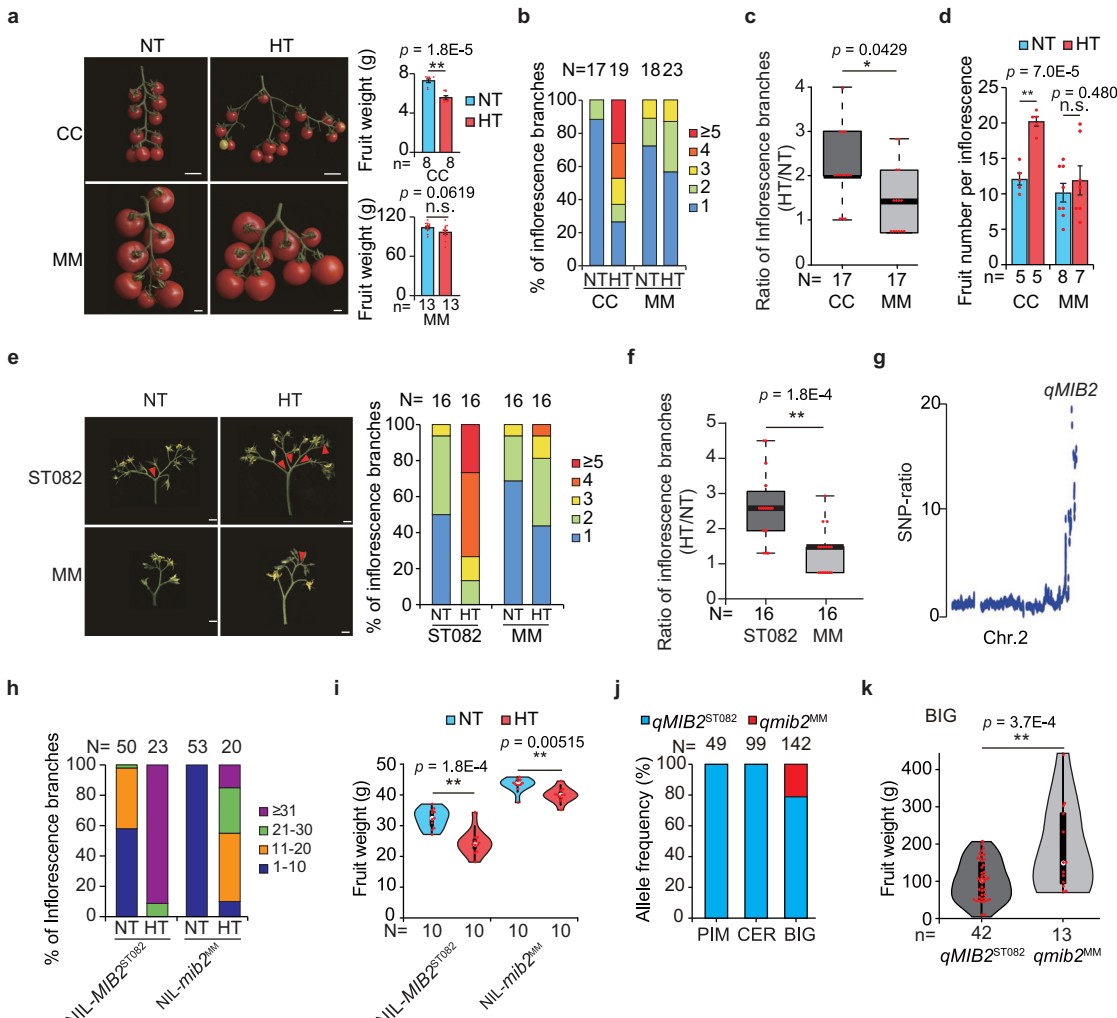

**Fig. 1 | Identification and characterization of *qMIB2*. a** Fruit phenotypes (left) and fruit weight (right) of *Solanum lycopersicum* var. *cerasiforme* LA1310 (CC) and *S. lycopersicum* Moneymaker (MM) under normal temperature (NT) and high temperature (HT) conditions in the field. Scale bars, 1 cm. **b** Inflorescence branch number of CC and MM under NT and HT conditions. **c** Ratio of inflorescence branch number for CC and MM plants between HT and NT conditions. **d** Fruit number per truss of CC and MM plants under NT and HT conditions. **e** Inflorescence phenotypes (left) and quantification (right) of inflorescence branch number of ST082 and MM plants. Scale bars, 1 cm. **f** Ratio of inflorescence branch number of ST082 and MM plants between HT and NT conditions. **g** Detailed view of the SNP ratio on chromosome 2. **h** Inflorescence branch number of near-isogenic lines (NILs) under NT and HT conditions. NIL-*MIB2*^ST082 and NIL-*mib2*^MM harbor the ST082 and Moneymaker (MM) genotypes at *MIB2*, respectively. **i** Fruit weight of the NILs under NT and HT conditions. **j** Allele frequencies of *qMIB2* in accessions classified as *S. pimpinellifolium* accessions (PIM), *S. lycopersicum* var. *cerasiforme* accessions (CER), and big-fruited *S. lycopersicum* accessions (BIG). N, number of accessions. **k** Fruit weight of big-fruited *S. lycopersicum* accessions with *qMIB2*^ST082 or *qmib2*^MM. Data in (**a, c, d, f, i, k**) were compared by two-tailed Student's *t*-test, **, *p* < 0.01; n.s. no significant difference (*p* > 0.05). Values in (**a, d**) are means ± SEM. Box edges in (**c, f, i, k**) represent the 0.25 and 0.75 quantiles, and the bold lines indicate median values. Whiskers indicate 1.5 times the interquartile range, the bar ranges the minimum to maximum observations. n in (**a, c, d, f, i, k**) = number of plants. N in (**b, e, h, j**) = number of inflorescences. Source data are provided as a Source Data file.

## Identification of target genes of MIB2

We established that MIB2 activates transcription in yeast (*Saccharomyces cerevisiae*) and localizes to the nucleus, as might be expected for a transcription factor (Supplementary Fig. 3f, g). To explore how MIB2 regulates inflorescence branching, we performed chromatin immunoprecipitation with an anti-HA antibody followed by sequencing (ChIP-seq) to identify all MIB2 binding sites using meristematic tissues of *proMIB2:MIB2-YFP-HA* complementation plants. We identified 4,785 overlapping peaks around 4,100 adjacent genes (Supplementary Fig. 4a, Supplementary Data 1). Among these, approximately 57% occurred in proximal promoter regions (< 5 kb upstream of the ATG) (Fig. 3a). The most significantly enriched motif was CACGTG (E = 3.6e − 079) (Fig. 3b), a conserved G-box motif recognized by bHLH transcription factors[23], suggesting that MIB2 directly binds to its target genes by recognizing the G-box elements in their promoters. Many of

MIB2 binding sites were distributed around transcription start sites (Supplementary Fig. 4b), suggesting that MIB2 binds to promoter regions to regulate transcription. To confirm the MIB2 target genes, we performed transcriptome deep sequencing (RNA-seq) analysis using meristematic tissues of NIL-*mib2*^MM and NIL-*MIB2*^ST082 plants. Compared to NIL-*MIB2*^ST082 meristems, 231 genes were downregulated and 131 genes were upregulated in NIL-*mib2*^MM meristems (Supplementary Fig. 4c, Supplementary Data 2). Among these, 88 downregulated and 37 upregulated genes overlapped with putative MIB2 target genes identified by ChIP-seq (Fig. 3c).

Gene ontology (GO) term enrichment analysis of these target genes suggested that they are associated with responses to various stresses, especially heat (Supplementary Fig. 4d). Several *HEAT SHOCK PROTEIN* (*HSP*) and *HEAT SHOCK FACTOR* (*HSF*) genes were downregulated in NIL-*mib2*^MM meristems (Supplementary Data 2). MIB2

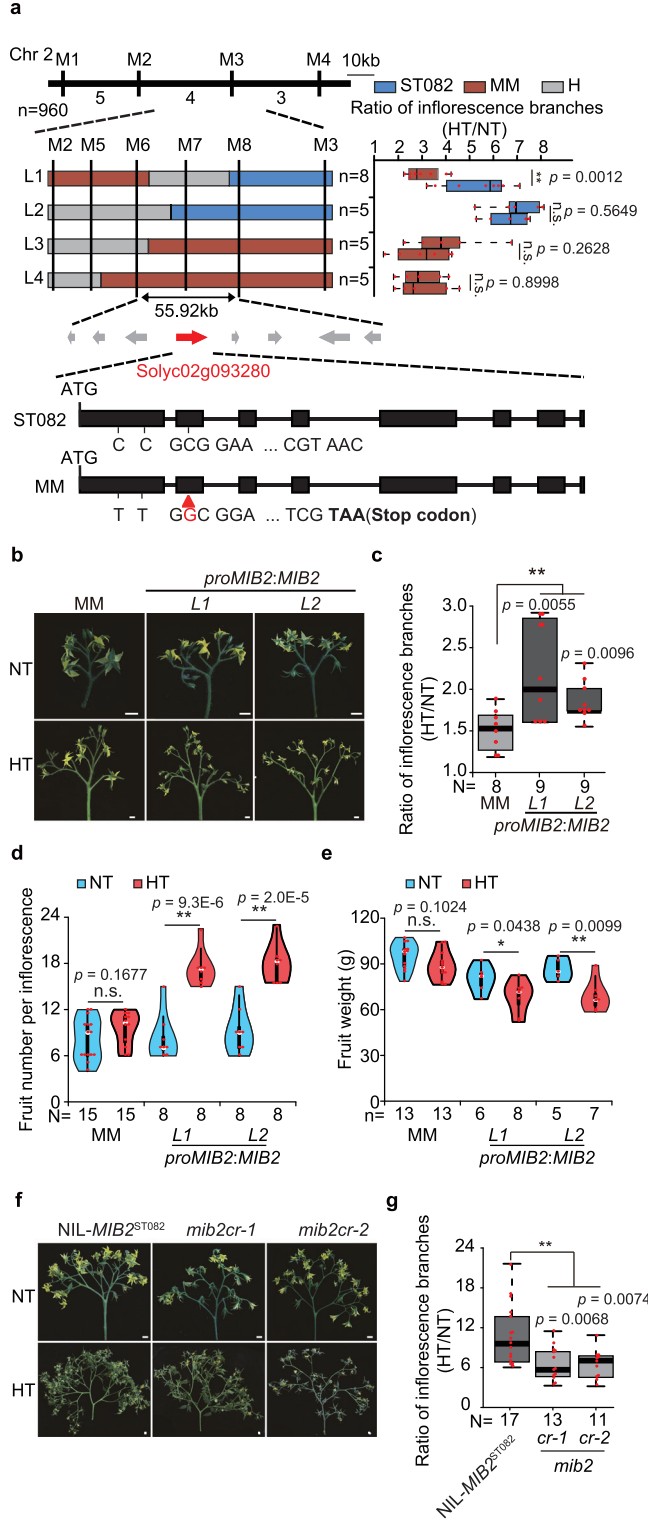

**Fig. 2 | Map-based cloning of qMIB2. a** Fine mapping of *qMIB2*. Top panel, positional cloning narrowing *qMIB2* to a DNA segment between markers M5 and M7. The numbers below the bars indicate the number of recombinants. Middle panel, high-resolution mapping of *qMIB2* (left), and progeny test of four recombinants (right). The blue, red, and gray boxes indicate chromosome regions with the homozygous ST082, homozygous MM, and heterozygous genotypes, respectively. Bottom panel, diagram showing the candidate open reading frames. Two single nucleotide polymorphisms (SNPs) in the first exon cause two amino acid substitution, and a single nucleotide insertion in the second exon of *Solyc02g093280* in MM causes early translation termination. **b** Inflorescence phenotypes of MM and *proMIB2:MIB2-YFP-HA* complementation lines. Scale bars, 1 cm. **c** Ratio of inflorescence branch number of MM and *proMIB2:MIB2-YFP-HA* plants between HT and NT conditions. **d** Fruit number per inflorescence of MM and *proMIB2:MIB2-YFP-HA* plants under NT and HT conditions. **e** Fruit weight of MM and *proMIB2:MIB2-YFP-HA* plants under HT and NT conditions. **f** Inflorescence phenotypes of NIL-*MIB2*[ST082] and *mib2cr* plants. Scale bars, 1 cm. **g** Ratio of inflorescence branch numbers for NIL-*MIB2*[ST082] and *mib2cr* plants between HT and NT conditions. Data in (**a**, **c**, **d**, **e**, **g**) were compared by two-tailed Student's *t*-test, **, *p* < 0.01; *, *p* < 0.05; n.s. no significant difference (*p* > 0.05). Box edges in (**a**, **c–e**, **g**) represent the 0.25 and 0.75 quantiles, and the bold lines indicate median values. Whiskers indicate 1.5 times the interquartile range, the bar ranges the minimum to maximum observations. n in (**a**, **e**) = number of plants. N in (**c**, **d**, **g**) = number of inflorescences. Source data are provided as a Source Data file.

## MIB2 is a positive regulator of hypocotyl thermomorphogenesis

To explore how MIB2 responds to HT, we compared *MIB2* transcript levels in meristematic tissues at 22 °C and 35 °C. *MIB2* transcript levels increased at 4 hours and reached their highest level at 8 hours of 35 °C treatment (Fig. 4a). In line with its roles in transcriptional induction, MIB2 accumulated in *proMIB2:MIB2-YFP-HA* complementation plants at 4 hours and was highly enriched at 8 hours of 35 °C treatment (Fig. 4b). In addition, MIB2 accumulates at its own promoter regions to activate its transcription under HT treatment (Supplementary Fig. 4h–j). These data indicate that *MIB2* is induced by HT.

Elongated hypocotyls and accelerated flowering are hallmarks of thermomorphogenesis[24–26]. Moreover, the expression of *MIB2* in hypocotyls and cotyledons of NIL-*MIB2*[ST082] was also induced by high temperature (Supplementary Fig. 5a). Therefore, to investigate whether MIB2 mediates thermomorpho-genesis, we measured hypocotyl length in *proMIB2:MIB2-YFP-HA* and *mib2cr* seedlings grown at 22 °C or 35 °C. The hypocotyls were significantly elongated in *proMIB2:MIB2-YFP-HA* seedlings but were significantly shorter in *mib2cr* seedlings at 35 °C compared to the MM control. We observed no differences in hypocotyl length in *proMIB2:MIB2-YFP-HA* or *mib2cr* seedlings at 22 °C relative to MM (Fig. 4c, d). These results indicate that the mutation of *MIB2* resulted in insensitivity to HT in terms of hypocotyl elongation. By contrast, MM, *mib2cr*, and *proMIB2:MIB2-YFP-HA* plants flowered at the same time under NT and HT conditions (Supplementary Fig. 5b, c). Therefore, MIB2 is a HT-responsive regulator of seedling thermomorphogenesis in tomato.

### SlCOL1 is a target of MIB2 in inflorescence thermomorphogenesis

Since MIB2 regulates inflorescence architecture in response to HT, we identified the target genes of MIB2 in meristems based on ChIP-seq and RNA-seq data. *CONSTANS-LIKE 1* (*SlCOL1, Solyc02g089540*), whose *Arabidopsis* homolog is expressed in meristematic tissues to regulate flowering[27–29], was downregulated in NIL-*mib2*[MM] and bound by MIB2, suggesting that *SlCOL1* is a downstream target of MIB2 in tomato meristems. We confirmed that MIB2 binds to the P2 and P4 regions of the *SlCOL1* promoter by ChIP-qPCR using an independent ChIP sample (Fig. 5a). The interaction between recombinant maltose-binding protein (MBP)-MIB2 protein and an *SlCOL1* probe containing a G-box motif was attenuated by adding unlabeled competitor probe and abolished

accumulated at the promoter regions of *HSP90*, *HSP70*, and *HSFA6B* to activate their transcription (Fig. 3d–f, Supplementary Fig. 4e–g). Furthermore, *HSP90*, *HSP70*, and *HSFA6B* transcript levels were significantly higher in NIL-*MIB2*[ST082] than in NIL-*mib2*[MM] meristems under HT treatment (Fig. 4f, Supplementary Fig. 4g), indicating that MIB2 activates their transcription in response to HT. These results suggest that HT is an important cue for the role of MIB2 in meristem determination.

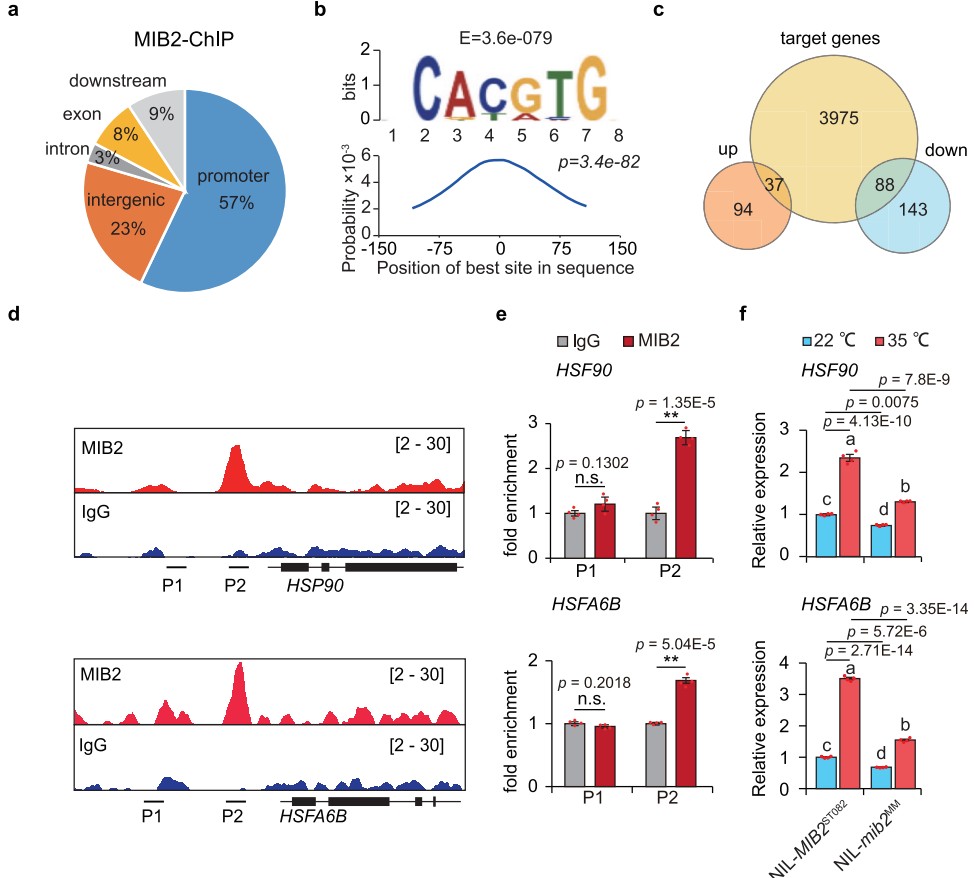

**Fig. 3 | Identification of MIB2 target genes. a** Pie chart displaying the genomic distribution of MIB2 binding sites in the meristematic tissues of *proMIB2:MIB2-HA* plants by ChIP-seq. Promoter, −5 kb to −1 bp upstream of the ATG; downstream region, +1 bp to +1 kb downstream of the stop codon. **b** The most significant MIB2 binding motif produced by MEME. The bottom graph shows the distribution of the motif corresponding to the −150 to +150 bp region flanking the binding-site peaks. The E-value of a motif is based on its log likelihood ratio. The x-axis indicates the position of the binding motif in the sequence, and the y-axis indicates the probability of the identified motif. Each "motif probability curve" shows the (estimated) probability of the best match to a given motif occurring at a given position in the input sequences. The *p*-value is calculated by using the one-tailed binomial test on the number of sequences with a match to the motif that have their best match in the reported region. **c** Venn diagram showing the number of overlapping genes between the putative MIB2-bound genes identified by ChIP-seq and differentially regulated genes in NIL-*mib2*^MM relative to NIL-*MIB2*^ST082 as identified by RNA-seq.

**d** Chromatin binding profiles of MIB2 at the *HEAT SHOCK PROTEIN 90* (*HSP90*) and *HEAT SHOCK TRANSCRIPTION FACTOR 6B* (*HSFA6B*) promoters. The short black lines labeled P1 and P2 represent the regions used for ChIP-qPCR. **e** ChIP-qPCR analysis of the bound regions in the *HSP90* and *HSFA6B* promoters. P1 and P2 represent the regions used for ChIP-qPCR. Input was used as a negative control. The DNA fragment of the *ACTIN* (*Solyc03g078400*) 3′ intergenic region was used as an internal control. Data were compared by two-tailed Student's *t*-test, **, *p* < 0.01; n.s. no significant difference (*p* > 0.05). Values are means ± SEM (n = 4 biologically independent replicates). **f** Expression levels of *HSP90* and *HSFA6B* in meristematic tissues of NIL-*MIB2*^ST082 and NIL-*mib2*^MM plants under NT and HT conditions. *UBIQUITIN3* (*Solyc01g056940*) was used as the reference transcript. Different letters indicate a significant difference (*p* < 0.05) based on the one-way ANOVA followed by Tukey's multiple comparisons test. Values are means ± SEM (n = 4 biologically independent replicates). Source data are provided as a Source Data file.

---

using a mutated G-box probe, confirming that MIB2 directly targets *SlCOL1* via the G-box motif (Fig. 5b).

In a dual firefly luciferase (LUC) reporter assay in *Nicotiana benthamiana* leaves, *SlCOL1* transcriptional output doubled when *proSlCOL1:LUC* was co-infiltrated with *pro35S:MIB2-FLAG* compared to the empty vector control (Fig. 5c), indicating that MIB2 activates *SlCOL1* transcription. Consistent with this finding, the expression levels of *SlCOL1* in the meristematic tissues were decreased in the NIL-*mib2*^MM and *mib2cr* mutants relative to NIL-*MIB2*^ST082 and increased in *proMIB2:MIB2-YFP-HA* plants relative to MM (Fig. 5d, e and Supplementary Fig. 5d). Like *MIB2*, *SlCOL1* was also expressed in various meristems and highly expressed in floral meristems and sympodial inflorescence meristems (Supplementary Fig. 5e). These data reveal that *SlCOL1* is directly bound and activated by MIB2 in tomato meristems.

Given the striking effect of ambient temperature on *MIB2* expression, we reasoned that the role of MIB2 in regulating *SlCOL1* transcription might also be temperature-dependent. Indeed, ChIP-

qPCR revealed that MIB2 binding to the *SlCOL1* promoter is strongly temperature-dependent, with an approximately six-fold and ten-fold increase at the P2 and P4 regions, respectively, after 8 hours of 35 °C treatment (Fig. 5f). With more MIB2 protein enriched at the *SlCOL1* promoter, *SlCOL1* transcript levels increased under HT treatment. This inducible *SlCOL1* expression was abolished in the *mib2cr* mutants (Fig. 5g), indicating that MIB2 regulates *SlCOL1* transcription in a temperature-dependent manner.

To further investigate the roles of SlCOL1 in inflorescence development as a MIB2 target, we overexpressed *SlCOL1* in NIL-*mib2*^MM plants. Two *pro35S:SlCOL1-YFP-HA* overexpression lines exhibited more inflorescence branching than NIL-*mib2*^MM plants (Fig. 5h, Supplementary Fig. 5f), indicating that overexpressing *SlCOL1* partially rescued the inflorescence phenotypes resulting from the mutation of *MIB2*. Therefore, overexpressing *SlCOL1* helps overcome the requirement for MIB2 in HT-responsive inflorescence branching.

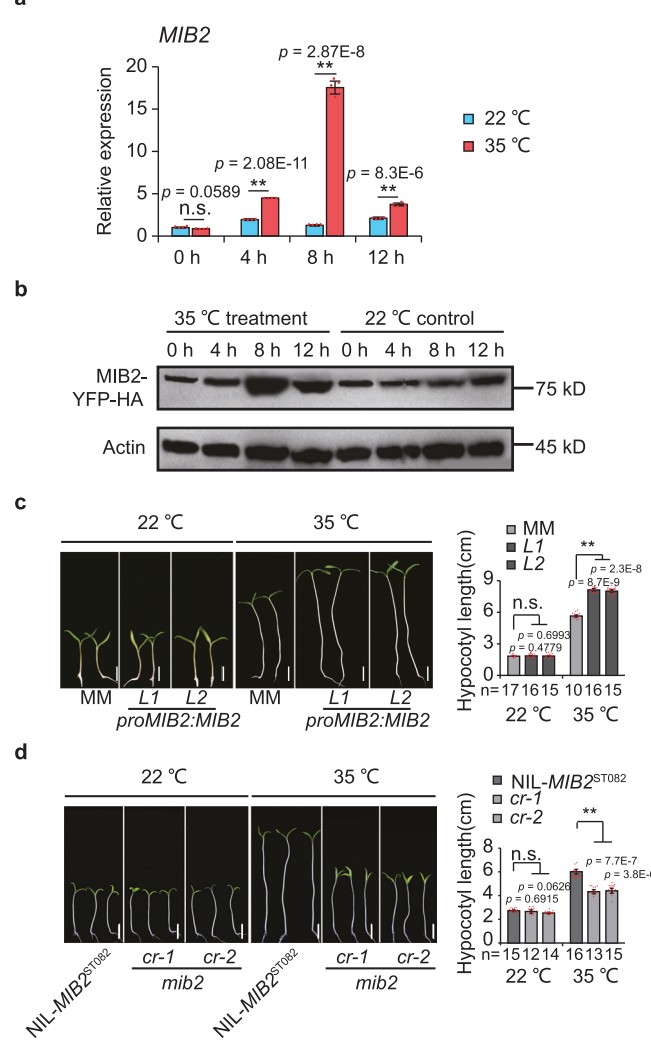

**Fig. 4 | Regulation of *MIB2* by high temperature. a** Relative *MIB2* expression in the meristematic tissues of NIL-*MIB2*[ST082] during a transition from 22 °C to 35 °C. Seedlings were grown to the 4-leaf stage at 22 °C in moderate sunlight, transferred to 22 °C in full sunlight and incubated for 5 days, and transferred to 35 °C in full sunlight conditions. Samples were collected and analyzed at the indicated time points. Seedlings grown at 22 °C were used as a control. *UBIQUITIN3* (*Solyc01g056940*) was used as the reference transcript. Data were compared by two-tailed Student's *t*-test, **, *p* < 0.01; n.s. no significant difference (*p* > 0.05). Values are means ± SEM (n = 4 biologically independent replicates). **b** Immunoblot analysis of MIB2 abundance in *proMIB2:MIB2-YFP-HA* meristematic tissues during the transition from 22 °C to 35 °C using anti-HA (Sigma-Aldrich, H6908, 1:2000). Seedlings were grown to the 4-leaf stage at 22 °C under moderate sunlight, transferred to 22 °C under full sunlight and incubated for 5 days, and transferred to 35 °C under full sunlight conditions. Samples were collected and analyzed at the indicated time points. Seedlings grown at 22 °C were used as a control. The anti-actin (Sigma-Aldrich, A0480, 1:2000) was used as a loading control. **c, d** Hypocotyl lengths of MM and *proMIB2: MIB2-YFP-HA* complementation seedlings (**c**) and NIL-*MIB2*[ST082] and *mib2cr* seedlings (**d**) grown at 22 °C or 35 °C under a short-day photoperiod. Seedlings were grown for 7 days at 22 °C or for 3 days at 22 °C, transferred to 35 °C, and grown for an additional 4 days before being photographed. Representative images of seedlings (left). and quantification of hypocotyl length (right). Scale bars, 1 cm. Data were compared by two-tailed Student's *t*-test, **, *p* < 0.01; n.s. no significant difference (*p* > 0.05). Values are means ± SEM. n in (**c, d**) = biologically independent samples. Source data are provided as a Source Data file.

## Discussion

Our work reveals the roles of a bHLH transcription factor MIB2 in regulating inflorescence plasticity in response to HT. Unlike its ortholog SPT in Arabidopsis, which regulates seed germination and vegetative growth in response to cold stress[19,20,22], MIB2 accumulates in response to elevated ambient temperature and positively regulates hypocotyl elongation in tomato, suggesting that its functions are similar to those of PHYTOCHROME-INTERACTING FACTOR4 (PIF4), a central regulator of hypocotyl thermomorphogenesis[30–32]. However, unlike the roles of PIF4 in flowering[33], MIB2 promotes inflorescence branching but does not regulate flowering in tomato, although its encoding gene is expressed in all meristems, and overexpressing its target gene *SlCOL1* delayed flowering (Supplementary Fig. 5g). Our results indicate that MIB2 specifically regulates inflorescence development in tomato meristems.

Disruption of MIB2 enhances tomato inflorescence stability that results in no significant change of MM inflorescence branches in responsive to HT (Fig. 1f). It is worthy to note that the inflorescence branches of *mib2cr* mutant in NIL-*MIB2*[ST082] are changed by about 6 times, which is more than 2.5-fold changes of inflorescence branching in ST082, although the variation of inflorescence branching is still less than that of the control NIL-*MIB2*[ST082] plants from NT to HT (Fig. 2g), suggesting that there are other factors in NIL-*MIB2*[ST082] regulate inflorescence thermomorphogenesis. The much more inflorescence branches of NIL-*MIB2*[ST082] than that of ST082 indicates that the NIL-*MIB2*[ST082] plants might have other alleles which contributes to tomato inflorescence branches. In fact, different with ST082 plants with the *j2* alleles, the NIL-*MIB2*[ST082] plants have the *j2* and *STM3*[D] alleles, which positively controls inflorescence branching[6,15,34], suggesting that STM3 may be the other factor functions in tomato inflorescence thermomorphogenesis.

Although MIB2 promotes inflorescence branching, which enhances the adaptability of tomato to elevated ambient temperatures, the *mib2* haplotype might be predominantly selected in some large-fruited tomato varieties. The decrease in fruit weight accompanied by the increase in inflorescence branching may partially explain why breeders intentionally select for *mib2*, especially in large-fruited varieties. Modern large-fruited tomatoes carrying the *mib2* haplotype with appropriate inflorescence branching effectively overcome the trade-off between fruit number and size, thereby ensuring higher or stable yields at high ambient temperatures (Supplementary Fig. 6). Our findings lay the foundation for breeding climate-resilient crops with improved yield stability to cope with elevated temperatures in light of global climate change.

## Methods

### Constructs and plant materials

The CRISPR/Cas9 vectors targeting sites in the *MIB2* exons were designed using the CRISPR-P v2.0 tool (http://cbi.hzau.edu.cn/CRISPR2/). The vectors were constructed as previously described[35]. In brief, primers containing four single guide RNAs (sgRNAs) and BsaI recognition sites were used to amplify the sgRNAX_U6-26t_SlU6p_sgRNAX fragment using the pCBC_DT1T2_SlU6p vector as a template, after which the fragments were purified and cloned into pTX041 at the BsaI sites. The plasmids were validated by Sanger sequencing and transformed into Agrobacterium (*Agrobacterium tumefaciens*) strain AGL1. The CRISPR/Cas9 vector was transformed into NIL-*MIB2*[ST082] by the leaf disc transformation method[36]. Genomic DNA was extracted from positive T₀ plants and used as a template for PCR. The fragment containing the target site was subjected to Sanger sequencing to identify homozygous genome-edited plants.

The *proMIB2:MIB2-YFP-HA* complementation vector was constructed using the native *MIB2* promoter (containing the 3.58-kb DNA sequence upstream of the ATG) and the full-length coding sequence of *MIB2*. The promoter fragment was amplified using ST082 genomic

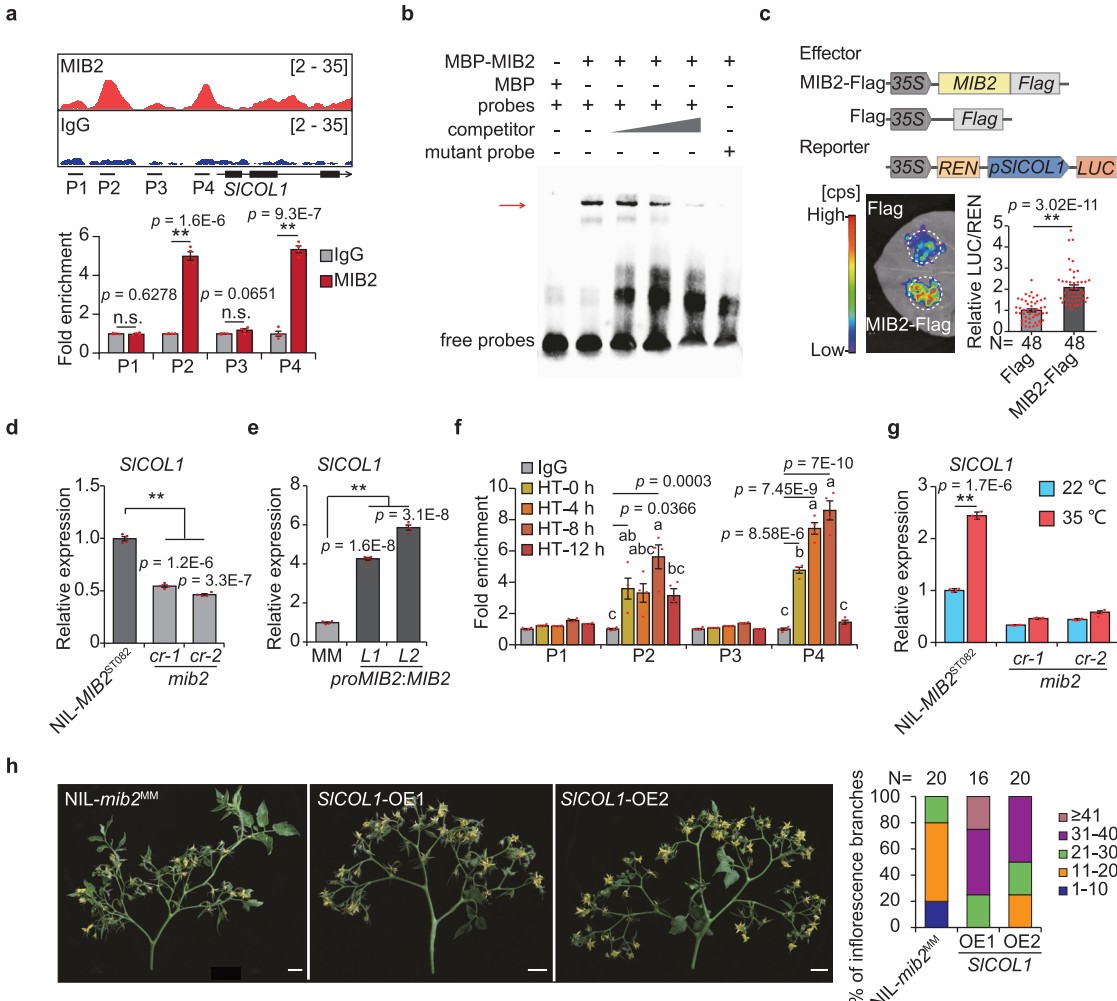

**Fig. 5 | *SlCOL1* is directly activated by MIB2. a** Chromatin binding profiles of MIB2, and ChIP-qPCR analysis of bound regions at the *SlCOL1* promoter by resampling. The short black lines labeled P1, P2, P3, and P4 represent the regions used for ChIP-qPCR. IgG was used as a negative control. The DNA fragment of the *ACTIN* (*Solyc03g078400*) 3′ intergenic region was used as an internal control. **b** Electrophoretic mobility shift assay (EMSA) of recombinant MBP-MIB2 protein with a biotin-labeled DNA fragment containing the G-box motif of the *SlCOL1* promoter. The arrow indicates the MIB2–DNA complex. Three independent experiments were performed. **c** Representative dual-luciferase reporter assay in *N. benthamiana* co-infiltrated with *pro35S:MIB2-FLAG* or *pro35S:FLAG* and *proSl-COL1:LUC-pro35S:REN*. Leaves co-infiltrated with *proSlCOL1: LUC-pro35S: REN* and *pro35S:FLAG* were used as a control. Renilla luciferase (REN) was used as an internal control. Data were compared by two-tailed Student's *t*-test, **, *p* < 0.01. Values are means ± SEM. N = biologically independent samples. **d, e** Relative *SlCOL1* expression in meristematic tissues of NIL-*MIB2*ST082 and *mib2cr* plants (**d**) and MM and *proMIB2:MIB2-YFP-HA* complementation plants (**e**). *UBIQUITIN3* (*Solyc01g056940*) was used as an internal control. **f** ChIP-qPCR analysis showing the enrichment of

MIB2 at the *SlCOL1* promoter after high temperature (35 °C) treatment. Meristematic tissues were collected from *proMIB2:MIB2-YFP-HA* transgenic plants and analyzed at the indicated time points. P1, P2, P3 and P4 represent the regions used for ChIP-qPCR. IgG was used as a negative control. **g** Relative *SlCOL1* expression in the meristematic tissues of NIL-*MIB2*ST082 and *mib2cr* plants after 8 hours of 35 °C treatments. Seedlings were grown to the 4-leaf stage at 22 °C under moderate sunlight, transferred to 22 °C in full sunlight and grown for 5 days, and transferred to 35 °C and grown for 8 hours in full sunlight. **h** Inflorescence phenotypes (left) and inflorescence branch number (right) of NIL-*mib2*MM and *SlCOL1* overexpression transgenic plants (*SlCOL1*-OEs) under HT conditions. N = inflorescence number. Scale bars, 3 cm. Data in (**a, d, e, g**) were compared by two-tailed Student's *t*-test, **, *p* < 0.01; n.s. no significant difference (*p* > 0.05). Values are means ± SEM (n = 4 biologically independent replicates). Different letters in (**f**) indicate a significant difference (*p* < 0.05) based on the one-way ANOVA followed by Tukey's multiple comparisons test. Values are means ± SEM (n = 4 biologically independent replicates). Source data are provided as a Source Data file.

DNA as a template, and the *MIB2* coding sequence was amplified using cDNA prepared from total RNA extracted from wild-type ST082 as a template. The promoter fragment was cloned into the vector pCAMBIA2300-YFP-HA using an In-Fusion cloning system kit (Clontech, 639648), and the *MIB2* coding sequence fragment was cloned into the resulting vector using T4 DNA ligase (TaKaRa, 2011B). The *proMIB2:MIB2-YFP-HA* complementation vector was validated by Sanger sequencing and transformed into Agrobacterium strain AGL1. The complementation vector was transformed into tomato cv. MM.

The *pro35S:MIB2-YFP-HA* and *pro35S:MIB2-FLAG* vectors were constructed using the full-length coding sequence of *MIB2*, which was

amplified using ST082 cDNA as a template and cloned into the *pCAMBIA2300-pro35S:YFP-HA* or *pCAMBIA2300-pro35S:FLAG* vector. The *pro35S:MIB2-YFP-HA* and *pro35S:MIB2-FLAG* vectors were validated by Sanger sequencing and transformed into Agrobacterium strain EHA105.

The *pro35S:SlCOL1-YFP-HA* vector was constructed using the full-length genomic region of *SlCOL1* (*Solyc02g089540*), which was amplified using ST082 genomic DNA as a template and cloned into the *pCAMBIA2300-pro35S:YFP-HA* vector. The *pro35S:SlCOL1-YFP-HA* vector was validated by Sanger sequencing and transformed into Agrobacterium strain AGL1. The plasmid was transformed into NIL-*mib2*MM,

and five independent $T_0$ transgenic plants were obtained. *SlCOL1*-OE1 and *SlCOL1*-OE2 are overexpression lines with higher *SlCOL1* expression than the control.

The NILs were generated by crossing and backcrossing between ST082 and MM. The offsprings were tested by molecular markers. After four generations, we examined the genetic identity by more than 100 markers (8-10 for each chromosome). At last, the genome similarity between the NILs was 98.02%. All primers used are listed in Supplementary Data 3.

## Growth conditions

For HT conditions, tomato (*Solanum lycopersicum*) plants were grown in a greenhouse in Beijing, China from May to August (HT, daily mean temperature: 32.2 °C, peak temperature: 38.7 °C, 12-h light/12-h dark photoperiod). For NT conditions, the plants were grown in a greenhouse in Zhangjiakou, Hebei Province, China from May to August (NT, daily mean temperature: 23.2 °C, peak temperature: 26.2 °C, 12-h light/12-h dark photoperiod). More than five individual plants per line or genotype were used to examine inflorescence branching and fruit weight; 5–8 inflorescences or fruits were evaluated per plant. The inflorescence branch number ratio (HT/NT) is the ratio of inflorescence branch number between HT and NT conditions. The fruit weight ratio (HT/NT) is the ratio for fruit weight between HT and NT conditions.

The seeds were surface-sterilized using 15% bleach, and then washed four times by ddH$_2$O, and stratified at 4 °C for 2 days. The seeds were then sowed on the 1/2 Murashige and Skoog (MS) medium and incubated at 22 °C for 3 days. All the seeds have been germinated and then the seedlings with the same hypocotyl length were selected and transferred into 35 °C. After 4 days, the hypocotyl lengths were measured.

## In situ hybridization

In situ hybridization was performed as described[13,37] with some modifications. Briefly, *MIB2* cDNA segments were amplified with the primers XP5795/XP5796 (Supplementary Data 3) and cloned into pEAZY-T3 (TransGen, CT301-01), which contains T7 promoter sequences. In vitro transcription was performed with T7 RNA polymerase to produce the antisense or sense probe for in situ hybridization. Meristem tissues were dissected and fixed for 24 h at 4 °C in freshly prepared 4% (w/v) paraformaldehyde buffered with phosphate-buffered saline (PBS, pH 7.2). The fixed tissues were dehydrated in graded ethanol: HistoChoice (Sigma, H2779) series and embedded in Paraplast Plus (Sigma, P3683). Dewaxed thin sections (10 µm) were hybridized with the probes for 12 h at 55 °C. Color development was observed in complete sections under a bright-field microscope (Olympus, BX43).

## Subcellular localization

The *pro3SS:MIB2-YFP-HA* vector was constructed with the coding sequence of *MIB2* using ST082 cDNA as a template and cloned in-frame and upstream of the sequence encoding YFP-HA. The resulting plasmid was transformed into Agrobacterium strain EHA105, which was co-infiltrated into the leaves of four-week-old *Nicotiana benthamiana* plants together with the RNA silencing suppressor P19. Protoplasts were isolated from 10-day-old *proMIB2:MIB2-YFP-HA* complementation seedlings. The shoots were cut into approximately 0.5-mm strips and incubated in 0.6 M D-mannitol (Sigma #78513) solution for 10 min, 0.3 g Cellulase and 0.08 g macerozyme (Yakult #R-10 and RS) were added to the 20 ml enzyme digestion buffer, followed by enzymatic digestion in the dark with gentle shaking for 5 h. The protoplasts were collected by filtration through 100-µm nylon filters (Thermo Fisher #352360). After washing with W5 buffer, the protoplasts were diluted in MMG buffer. YFP fluorescent signals were observed under a confocal laser-scanning microscope (Leica DM6 CS). A HFT514 dichroic mirror and a BP525-570 IR emission filter were used.

## Transcriptional activity assay in yeast

The full-length coding sequence of *MIB2* was cloned into the bait vector pGBKT7 using ST082 cDNA as a template. The pGBKT7-MIB2 and pGADT7 plasmids were co-transformed into yeast (*Saccharomyces cerevisiae*) strain AH109 according to the Clontech Yeast Protocols Handbook (TaKaRa, PT3024-1). The resulting positive colonies were obtained by growth on synthetic defined (SD) medium lacking leucine and tryptophan, and restreaked on plates containing SD medium lacking leucine and tryptophan but containing X-a-Gal for 3 days at 30 °C.

## Tissue collection, RNA extraction, and gene expression analysis

Meristems from approximately four-week-old plants were dissected within 2 hours of collection and immediately fixed in 100% acetone, followed by vacuum infiltration (0.06 MPa for 40 min)[38]. Approximately 60 sympodial inflorescence meristem and floral meristem (SIM and FM) tissues were microdissected under a stereomicroscope using tweezers, dried for 3 min at room temperature to remove the remaining acetone, and immediately frozen in liquid nitrogen. The materials were thoroughly ground in an MM300 mixer mill (Retsch) with a tungsten bead (3-mm diameter). Total RNA was extracted from each sample using a PicoPure RNA Extraction kit (Thermo-Fisher, 12204-01). Gene expression analysis was performed as the previously reported method with some modifications[39]. In brief, reverse transcription was performed with TransScript® II One-Step gDNA Removal and cDNA Synthesis SuperMix (TransGen, AH311-02) using 100 ng of total RNA. Quantitative PCR (qPCR) was performed with Taq Pro Universal SYBR qPCR Master Mix (Vazyme, Q712-02) on a Bio-Rad CFX-96 Real-Time PCR instrument using the following program: 3 min at 95 °C followed by 40 cycles of 20 s at 95 °C, 30 s at 60 °C, and 20 s at 72 °C. *UBI3* (Solyc01g056840) was used as the reference transcript for RT-qPCR. The primers used for RT-qPCR are listed in Supplementary Data 3.

## RNA-seq analysis

Total RNA was isolated from reproductive meristematic tissues collected from NIL-*MIB2*$^{ST082}$ and NIL-*mib2*$^{MM}$ according to the method as above for RT-qPCR. Three biological replicates were performed; each biological replicate contained at least 60 individuals. Six RNA-seq libraries were constructed and sequenced using an Illumina HiSeq2000 instrument at Berry Genomics (http://www.berrygenomics.com/). The filtered clean paired-end 150 bp reads were aligned to the tomato reference genome (ITAG 4.0) by STAR v2.5.3, and their features were counted using featureCounts v 1.5.3[40]. The statistical package DEGseq with the MA-plot-based method in R version 3.0.3 was used to calculate $p$-values, which were adjusted using the Benjamini-Hochberg procedure. The fold-change between NIL-*MIB2*$^{ST082}$ and NIL-*mib2*$^{MM}$ libraries was calculated using FPKM (fragments per kilobase of transcript sequence per million base pairs sequenced) as input data. The thresholds for the identification of differentially expressed genes (DEGs) were as follows: FPKM > 1 in at least one tissue, fold-change > 1.5 or <0.67, and Benjamini-Hochberg adjusted $p$-value (*padj*) < 0.05. Gene ontology (GO) term enrichment analysis of the DEGs was performed using the closest homologous genes in Arabidopsis with DAVID (The Database for Annotation, Visualization, and Integrated Discovery, https://david.ncifcrf.gov/).

## ChIP-seq and ChIP-qPCR

ChIP was performed using 0.5 g hand-dissected meristematic tissues of *proMIB2:MIB2-YFP-HA* transgenic plants with minor modifications and immediately frozen in liquid nitrogen. Meristems were completely ground in liquid nitrogen and crosslinked in 1% (w/v) formaldehyde (Sigma-Aldrich, 104003) for 10 min at 4 °C. The chromatin was sheared using a Diagenode Bioruptor Plus instrument to obtain ~300-bp DNA fragments. Immunoprecipitation was performed using an anti-

HA antibody (Sigma, H6908, 5 μg/sample). The DNA isolated by ChIP was used for qPCR analysis or Illumina-based paired-end sequencing.

ChIP-seq libraries were prepared using a NEXTflex Rapid DNA-seq kit for Illumina (NOVA, 5144-08) according to the manufacturer's protocol. The 150-bp single-end reads were mapped to the tomato reference genome (SL4.0) using Bowtie2 (version 2.3.5; parameters: -p 10 −5 15 −3 85 -phred 33)[41]. Duplicated and low-quality mapped reads were identified and removed by SAMtools (SAMtools view -F 1028 -Sb -q 10)[42]. Enriched bound peaks were then identified using MACS2 v.2.1[43]. The thresholds for peak calling were as follows: fold enrichment > 4, $q$-value < 0.00001. The overlapping peaks in two biological replicates were identified using a custom Perl script. MEME-ChIP and Homer were used for motif discovery and peak summits ($\pm$150 bp)[44]. For normalization and visualization, the filtered and sorted.bam files were converted to bigwig format using the bamCoverage function in deepTools v3.3.0[45]. ChIP-qPCR was performed using Taq Pro Universal SYBR qPCR Master Mix (Vazyme, Q712-02) on a Bio-Rad CFX-96 Real-Time PCR instrument with the following program: 3 min at 95 °C followed by 50 cycles of 20 s at 95 °C, 30 s at 60 °C, and 20 s at 72 °C. IgG was used as a negative control. A DNA fragment of the *ACTIN* (Solyc03g078400) 3′ intergenic region was used as an internal control. Primers used for qPCR are listed in Supplementary Data 3.

### Electrophoretic mobility shift assay

To construct the MBP-MIB2 plasmid, the full-length coding sequence of *MIB2* was amplified and ligated into the MBP-pMCSG7 plasmid containing a polyhistidine (6×His) sequence[46]. The recombinant protein was produced and purified from *Escherichia coli* strain BL21. The production of recombinant MBP-MIB2 and MBP was induced by incubation in 0.2 mM isopropylthio-β-D-galactoside (IPTG) at 16 °C for 16 h. The proteins were purified using Ni-NTA agarose (QIAGEN, 1018244) according to the manufacturer's instructions. The 5′ biotin-labeled DNA probes were synthesized by BGI (The Beijing Genomics Institute) and annealed. DNA gel shift assays were performed using a LightShift Chemiluminescent EMSA kit (Thermo-Fisher, 20148). Each EMSA reaction (20 μL) contained 1 μg purified recombinant protein, 4 μL biotin-labeled DNA probe (100 pmol), 2 μL binding buffer, and 1 μL poly (dI-dC). Transferred DNA and purified proteins were cross-linked using a UV lamp at 312 nm wavelength. The biotin-labeled DNA was detected using a Thermo Scientific chemiluminescence kit and visualized on a Tanon-5200 Chemiluminescent Imaging System (Tanon Science and Technology). The primers used to generate the constructs are listed in Supplementary Data 3.

### Dual luciferase reporter assay

For plasmid construction, the *proSlCOL1:LUC-pro35S:REN* backbone vector was obtained from pPZP211. The >3.0-kb promoter sequence of *SlCOL1* was amplified using ST082 genomic DNA as a template and integrated into *LUC-pro35S:REN* using an In-fusion HD Cloning kit (Clontech, 639649). The *pro35S:MIB2-FLAG* vector was constructed using the full-length coding sequence of *MIB2*, which was amplified using ST082 cDNA as a template and cloned into the *pCAMBIA2300-pro35S:FLAG* vector. The *pro35S:MIB2-FLAG* vector was validated by Sanger sequencing. All plasmids were transformed into Agrobacterium EHA105 competent cells. A single colony was cultured in LB medium until the OD$_{600}$ value reached 1. The Agrobacterium cells were collected by centrifugation (3214 x $g$, 10 min, 20 °C) and resuspended in 10 mM MgCl$_2$ and 150 μM acetosyringone until the OD$_{600}$ value reached 1. Cells containing the over-expression plasmids, luciferase plasmid, and P19 plasmid were mixed in a ratio of 2:1:1 (v/v/v) and infiltrated into *N. benthamiana* leaves using a syringe. The leaves were harvested and ground in liquid nitrogen 36 hours after infiltration. Firefly luciferase and Renilla luciferase activities were measured using a dual-luciferase reporter assay system (Promega; E1910) on a Promega GLOMAX 20/

20 LUMINOMETER. Renilla luciferase activities was used as an internal control. Thirty repeats were collected. Primers used to generate the constructs are listed in Supplementary Data 3.

### Reporting summary

Further information on research design is available in the Nature Portfolio Reporting Summary linked to this article.

## Data availability

The RNA-seq and ChIP-seq data generated in this study have been deposited in the NCBI Sequence Read Archive (SRA) database under accession PRJNA943168 and SRR12756256, respectively. Source data are provided with this paper.

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

## Acknowledgements
This work was supported by the National Natural Science Foundation of China Grant (31930099 to X. Cui), the Projects of International Cooperation and Exchanges NSFC (32120103010 to X. Cui), the National Key Research and Development Program of China (2022YFF1003002 to X. Cui), and the National Natural Science Foundation of China (32102403 to S. Sun). We thank Prof. Sanwen Huang (Chinese Academy of Agricultural Sciences), Hong Yu (Chinese Academy of Sciences), and Prof. Guosheng Xiong (Nanjing Agricultural University) for their revision of the manuscript.

## Author contributions
S.S. carried out most of the experiments; Z.L. analyzed the ChIP-seq data and RNA-seq data; X.W., S.F., J.S., J.K., and R.L. helped to observe the phenotypes; H.W. revised the paper. X.C. conceived the project and designed the research. S.S. and X.C. wrote the manuscript.

## Competing interests
The authors declare no competing interests.
