## [Peer Review File · Nature Communications]

Genetic control of thermomorphogenesis in tomato inflorescencesReviewers' Comments:

Reviewer #1:

Remarks to the Author:

This extensive and meticulous study investigates a QTL segregating in a population derived from a cross between cultivated tomato lines differing in the response of their inflorescence to high temperature. One line shows higher branching and smaller fruit, the other has the same branching and fruit size as plants raised at lower temperature. The QTL maps to a gene MIB2 encoding a transcription factor of the bHLH class, and loss of its function is associated with the loss of high temperature branching. By a series of sophisticated genetic, molecular and cellular methods, the authors provide evidence that MIB2 expression is boosted at high temperature and that it directly enhances expression of the SICOL1 gene, which in turn promotes inflorescence branching.

This is a novel and well documented study, with all steps clearly explained and conclusions carefully drawn. It is significant in that it provides a new role for a transcription factor not previously known to be involved in temperature response to inflorescence branching. Such a property could be useful in breeding new varieties of tomato fine-tuned to optimise fruit size and number depending on growth conditions.

There is little that needs specific comment. It is not necessary to carry out further experiments in my opinion.

1. One matter to note is the reliance on only two transgenic lines in several places. More would to strengthen the conclusion that effects are not insertion specific.
2. Another comment is necessary about the phylogeny of the MIB2 gene (Extended Data Figure 3). The neighbour joining method using full amino acid sequences has been long overtaken by maximum likelihood or Bayesian methods using concatenated conserved regions. Also, it needs to be explained briefly in the legend which group of bHLH members have been selected for analysis, especially since it includes PIF proteins.
3. An unfortunate mistake in lines 4-5 of the legend of Extended Data Figure 4 needs to be corrected. "...SICOL1 expression is sharply increased..." not reduced.

Reviewer #2:

Remarks to the Author:

The manuscript entitled "Genetic Control of Thermomorphogenesis in Tomato Inflorescences" demonstrates that the quantitative trait locus qMULTIPLE INFLORESCENCE BRANCH 2 (qMIB2) modulates inflorescence branching in response to high ambient temperatures in tomato (Fig. 1). Using a mapping population, the authors cloned the qMIB2 QTL and identified an insertion in the Solyc02g093280 (SPATULA) gene, leading to an early stop codon (mib2). Through elegant genetic analysis, including loss-of-function mutants and overexpression lines, they confirmed the function of the MIB2 gene in regulating inflorescence development (Fig. 2). The authors also demonstrated that MIB2 binds to and regulates the expression of SICOL1, thereby linking MIB2 to the regulation of inflorescence architecture (Fig. 5A). Collectively, the authors concluded that the MIB2-SICOL1 module plays a crucial role in enabling tomato inflorescences to adapt to high temperatures. Furthermore, they suggest that the non-functional mib2 allele was selected in large-fruited varieties to ensure uniform and larger fruits under varying temperatures.

Overall, the major results represent significant discoveries in the field of temperature signaling and development. The majority of the data presented in this manuscript supports the conclusion well and is backed by sufficient genetic evidence. The manuscript presents new and interesting data that is well worth being published in Nature Communications. I do not have many major concerns; however, some

comments and suggestions that may improve the manuscript are listed below:

Major:

1. In Fig. 2g, please present the absolute number of inflorescence branches. It seems that the increase in inflorescence branches is still very high in the mutant (6-fold HT/NT). In contrast, the change in ST082, for example, was only 2.5-fold. This suggests that other factors may regulate the number of inflorescence branches in response to HT. This aspect should be discussed.
2. Lines 160-171, I'm not convinced that MIB2 regulates hypocotyl elongation in response to high temperature. First, the seedlings were transferred to 35°C after three days in 22°C. At this point, the seeds are still not fully germinated, and since the germination rate can dramatically affect the hypocotyl length, it is not clear if MIB2 affects germination or hypocotyl elongation. This needs to be further examined. One option is to transfer to 35°C fully germinated seeds (minimum seven days at 22°C). Second, MIB2 is specifically expressed in the meristem (Extended Data Fig. 3a). Is it also expressed in the hypocotyl or the cotyledons? Does it make sense that MIB2 regulates hypocotyl elongation even though it is not expressed in these tissues?
3. Fig. 5d, the expression of SICOL1 in MM vs. LA1310 (cc) and/or NIL-MIB2ST082 vs. NIL-mib2MM needs to be shown, to evaluate the regulation of MIB2 over SICOL1 in the tomato cultivars.

Minor:

4. As a general suggestion, I would recommend changing the gene name from MIB2 to SISPATULA (SISPT), as "MIB" sounds similar to "MYB" and may lead to potential confusion in the future.
5. Lines 45-46, "... we observed no significant changes in fruit weight or inflorescence branching..." There are no statistics on the inflorescence branching between NT and HT.
6. It would be more informative to add to Extended Data Fig. 1b which RIL carries the MM allele and which carries the ST082 allele.
7. In Line 61, to generate the BC3F2, used in the bulk segregate analysis, to which background it was backcrossed. Please describe it in the methods section.
8. In lines 63-64, which markers were used during the backcrosses to generate the NIL lines? Please describe it in the methods section.
9. Line 88, "Inflorescence branching and fruit number were insensitive to HT..." While the Inflorescence branching was indeed less affected by the HT in qmib2MM in comparison to qMIB2ST082, there is a clear increase in the inflorescence branches also in qMIB2MM (Extended Data Fig. 2a). Therefore, I think the term less sensitive, rather the insensitive to HT will describe it better.
10. In Fig. 2a, according to the primer position, the markers M4, M5, M6, M7 in the second row need to be changed to M5, M6, M7, M8. Please check.
11. Extended Data Fig. 3c, please present the change in the protein sequence along with the difference in the DNA sequence.
12. Extended Data Fig. 3e, very low resolution; therefore, it was impossible to interpret the results.
13. In the ChIP-seq experiment, the IgG control is missing in the SRA database. In addition, since MIB2-YFP-HA plants were used, a control using anti-HA on non-transgenic plants would be better as it accounts for the nonspecific binding of the HA antibody.
14. Line 441, add that the Chip-seq data is also deposited.
15. Line 174, please add the Solyc number for SICOL1 and the same in the method section.

Reviewer #3:

Remarks to the Author:

The work reports genetic analysis of temperature-responsive inflorescence branching in tomato. The authors identified a major quantitative trait locus MIB2 controlling inflorescence branching in response to high growth temperature. MIB2 encodes a bHLH transcription factor orthologous to Arabidopsis SPATULA and its loss-of-function mutation as detected in Moneymaker and some other cultivated tomato eliminates the potential of inflorescence branching when plants are grown under relatively high temperature. Using ChIP-seq and RNA-seq technology, the authors identified putative MIB2 target

genes and further showed that MIB2 bound to the promoter region of SICOL1, a CONSTANS-like gene, to regulate its expression in response to high temperature. Based on the increased inflorescence branching in SICOL1 overexpression in *mib2* -- NILs harboring MoneyMaker (MM) allele of the MIB2 locus, the authors conclude that MIB2-SICOL1 module play an important role in regulation of tomato inflorescence plasticity in response to temperature. It will be an interesting story after some key issues are solved, particularly those concerning the timing of MIB2 and SICOL1 action on inflorescence development in response to temperature.

1. Solyc02g093280/bHLH017 was identified as qMIB2 candidate; the MoneyMaker genome contains an insertion of the nucleotide G in its second exon disrupted coding potential, leading premature translation. Given the reference Heinz1706 genome has a complete Solyc02g093280 and only small portion of BIG accessions contain this insertion, the MM allele of qMIB2 was unlikely selected during tomato breeding. More data are needed to support the conclusion made in this study that MIB2 was targeted by tomato breeders.

2. The field experiments concerning inflorescence branching under normal and high temperature (NT and HT) were conducted in two separate greenhouses located in two different places. Except temperature, other factors including soil, humidity and irrigation were varied between the two greenhouses. It is arguable that these factors or their interactions also had significant contribution to the observed variations in inflorescence branching. New experimental data collecting from well-controlled growth conditions are required to demonstrate that temperature is the key factor determining this inflorescence plasticity.

3. The weedy *S. lycopersicum* var. *cerasiforme* (cherry tomato) is generally thought the admixture of domesticated tomato and its wild ancestor *S. pimpinellifolium*; its origin likely predates domestication (Ranc et al BMC Plant Biol 2008,8:130; Razifard et al Mol Biol Evol 2020,37:1118-1132). It is incorrect to call it "wild cherry tomato" (Line 38).

4. The qMIB2 NILs were obtained by crossing RIL ST082 to MoneyMaker three times. Such NILs are not genetically pure enough for functional analysis of a particular trait locus, they still contained quite significant amount of genetic variations, affecting trait formation. Did the authors verify the genetic background of these NILs using DNA markers or genome survey by re-sequencing?

5. The inflorescence of NIL-*mib2*MM showing in Figure 5h are highly branched, very different when comparing with the MM inflorescences showing in Figure 1a and 2b. Please clarify the discrepancy.

6. Figure 3d-f presented ChIP-qPCR data to confirm MIB2 binding in HSP90 and HSFA6B promoter regions. These data are not relevant to the question the authors want to address -- regulation of inflorescence development by MIB2, if the authors can't provide data to show the two genes are part of MIB2 regulatory network. These data may go to supplemental.

7. Under what growth conditions the two SICOL1 overexpression lines showed inflorescence branching phenotype, NT or HT? Though SICOL1 overexpression was sufficient to increase inflorescence branching, it is more important to know whether SICOL1 is required for this trait. Inflorescence branching phenotypes of *slcol1* mutants in MIB2CC background will determine the significance of SICOL1 as MIB2 target in inflorescence branching.

8. The developmental stages of shoot meristems should be defined more specifically. The authors often just refer to meristems or reproductive meristematic tissues. It is unclear when MIB2 and SICOL1 take into play during inflorescence development.

Dear Reviewers,

We sincerely thank the editor and all reviewers for their positive and constructive comments that help us to improve our work. In this new version, we have been added some results to support our conclusion and carefully revised the main text.

Point-by-point responses to Reviewers' comments

Reviewer #1 (Remarks to the Author):

This extensive and meticulous study investigates a QTL segregating in a population derived from a cross between cultivated tomato lines differing in the response of their inflorescence to high temperature. One line shows higher branching and smaller fruit, the other has the same branching and fruit size as plants raised at lower temperature. The QTL maps to a gene *MIB2* encoding a transcription factor of the bHLH class, and loss of its function is associated with the loss of high temperature branching. By a series of sophisticated genetic, molecular and cellular methods, the authors provide evidence that *MIB2* expression is boosted at high temperature and that it directly enhances expression of the *SICOL1* gene, which in turn promotes inflorescence branching.

This is a novel and well documented study, with all steps clearly explained and conclusions carefully drawn. It is significant in that it provides a new role for a transcription factor not previously known to be involved in temperature response to inflorescence branching. Such a property could be useful in breeding new varieties of tomato fine-tuned to optimise fruit size and number depending on growth conditions.

Response: Thanks for your concise summary and the comments on our work.

There is little that needs specific comment. It is not necessary to carry out further experiments in my opinion.

1. One matter to note is the reliance on only two transgenic lines in several places. More would to strengthen the conclusion that effects are not insertion specific.

Response: Thank you to point it out. We have checked the insertion positions of the complementation lines and OE lines by sequencing and found that the T-DNAs are inserted at different sites. And the different gene expression levels in the *SICOL1*-OE transgenic lines (Extended Data Fig. 5e) indicate that they are independent lines.

2. Another comment is necessary about the phylogeny of the *MIB2* gene (Extended Data Figure 3). The neighbour joining method using full amino acid sequences has been long overtaken by maximum likelihood or Bayesian methods using concatenated conserved regions. Also, it needs to be explained briefly in the legend which group of bHLH members have been selected for analysis, especially since it includes PIF proteins.

Response: We gratefully appreciate your valuable suggestion and comment. We used maximum likelihood and Bayesian methods to construct the phylogenetic tree of *MIB2* and obtained the same results that *MIB2* is the closest homolog with Arabidopsis *SPT* (Extended Data Figure 3a). We have been revised the legend and indicated that the phylogenetic tree is about the PIFs subfamily of bHLH

transcription factor^{1,2}.

3. An unfortunate mistake in lines 4-5 of the legend of Extended Data Figure 6 needs to be corrected. "...SICOL1 expression is sharply increased..." not reduced.

Response: We are sorry for the careless mistake. We revised it in the new version.

Reviewer #2 (Remarks to the Author):

The manuscript entitled "Genetic Control of Thermomorphogenesis in Tomato Inflorescences" demonstrates that the quantitative trait locus *qMULTIPLE INFLORESCENCE BRANCH 2* (*qMIB2*) modulates inflorescence branching in response to high ambient temperatures in tomato (Fig. 1). Using a mapping population, the authors cloned the *qMIB2* QTL and identified an insertion in the *Solyc02g093280* (*SPATULA*) gene, leading to an early stop codon (*mib2*). Through elegant genetic analysis, including loss-of-function mutants and overexpression lines, they confirmed the function of the *MIB2* gene in regulating inflorescence development (Fig. 2). The authors also demonstrated that *MIB2* binds to and regulates the expression of *SICOL1*, thereby linking *MIB2* to the regulation of inflorescence architecture (Fig. 5A). Collectively, the authors concluded that the *MIB2-SICOL1* module plays a crucial role in enabling tomato inflorescences to adapt to high temperatures. Furthermore, they suggest that the non-functional *mib2* allele was selected in large-fruited varieties to ensure uniform and larger fruits under varying temperatures.

Overall, the major results represent significant discoveries in the field of temperature signaling and development. The majority of the data presented in this manuscript supports the conclusion well and is backed by sufficient genetic evidence. The manuscript presents new and interesting data that is well worth being published in *Nature Communications*. I do not have many major concerns; however, some comments and suggestions that may improve the manuscript are listed below:

Response: Thanks for your nice comments and suggestions. We have been revised our manuscript carefully and some results have been added in the new version.

Major:

1. In Fig. 2g, please present the absolute number of inflorescence branches. It seems that the increase in inflorescence branches is still very high in the mutant (6-fold HT/NT). In contrast, the change in ST082, for example, was only 2.5-fold. This suggests that other factors may regulate the number of inflorescence branches in response to HT. This aspect should be discussed.

Response: Thank you for the suggestion. We have added the absolute number of inflorescence branches of the NIL-*MIB2*^{ST082} and *mib2cr* mutants in Extended Data Fig. 3e. Moreover, we agree with your opinion about other factors in the NIL-*MIB2*^{ST082}, which affect the number of inflorescence branches in response to HT. The related discussion has been added in the main text.

2. Lines 160-171, I'm not convinced that *MIB2* regulates hypocotyl elongation in response to high temperature. First, the seedlings were transferred to 35°C after three days in 22°C. At this point, the seeds are still not fully germinated, and since the germination rate can dramatically affect the hypocotyl length, it is not clear if *MIB2* affects germination or hypocotyl elongation. This needs to be further examined. One option is to transfer to 35°C fully germinated seeds (minimum seven days at 22°C). Second, *MIB2* is specifically expressed in the meristem (Extended Data Fig. 3a). Is it also

expressed in the hypocotyl or the cotyledons? Does it make sense that MIB2 regulates hypocotyl elongation even though it is not expressed in these tissues?

Response: Thanks for your suggestion. To ensure seed germination, we sterilized the seeds with 15% bleach, and then washed and stratified at 4 °C for 2 days. The seeds are then put on the 1/2 MS medium and incubated at 22 °C for 3 days. All the seeds have been germinated and then the seedlings with the same hypocotyl length are selected and transferred into 35°C. After 4 days, we measured the hypocotyl lengths. The detailed information has been added in the methods.

According to your suggestion, we detected the expression of *MIB2* in the hypocotyl or the cotyledons at different temperature. The results showed that *MIB2* is expressed in the hypocotyl and cotyledons, and its expression level is higher at HT than at NT, suggesting that MIB2 might play its roles in regulation of hypocotyl elongation.

3. Fig. 5d, the expression of *SICOL1* in MM vs. LA1310 (cc) and/or NIL-*MIB2*^{ST082} vs. NIL-*mib2*^{MM} needs to be shown, to evaluate the regulation of MIB2 over *SICOL1* in the tomato cultivars.

Response: Thanks for your valuable suggestion. We detected the transcription levels of *SICOL1* in NIL-*MIB2*^{ST082} and NIL-*mib2*^{MM} under different ambient temperature by *qPCR* and the result showed that *SICOL1* expression is repressed in the NIL-*mib2*^{MM} and is more significantly repressed at HT. We have added the information in the Extended Data Fig. 5c. As you known, the genomic background of the MM vs. LA1310 (cc) are big difference, there are many other factors might affect *SICOL1* expression. Thus, we compared the transcription levels of *SICOL1* between the NIL-*MIB2*^{ST082} and NIL-*mib2*^{MM}.

Minor:

4. As a general suggestion, I would recommend changing the gene name from *MIB2* to *SISPATULA* (*SISPT*), as "MIB" sounds similar to "MYB" and may lead to potential confusion in the future.

Response: Thanks for your suggestion. Although the *SISPT* is indeed a reasonable name for this gene, the “MIB” (Multiple Inflorescence Branch) is for a series QTL mutants discovered in our lab. We prefer to use *MIB2* as this gene name for following our previously published *MIB1* gene.

5.Lines 45-46, "... we observed no significant changes in fruit weight or inflorescence branching..." There are no statistics on the inflorescence branching between NT and HT.

Response: Thanks. We have revised the description in this version. “we observed no significant changes in fruit weight and a slight increase in inflorescence branching”.

6. It would be more informative to add to Extended Data Fig. 1b which RIL carries the MM allele and which carries the ST082 allele.

Response: Thanks. According to your suggestion, we have added the information in the Extended Data Fig. 1b.

7. In Line 61, to generate the BC3F2, used in the bulk segregate analysis, to which background it was backcrossed. Please describe it in the methods section.

Response: Thank you for the comments. The detailed information has been added in the methods.

8. In lines 63-64, which markers were used during the backcrosses to generate the NIL lines? Please describe it in the methods section.

Response: Thank you for your comments. We have revised the methods.

9. Line 88, "Inflorescence branching and fruit number were insensitive to HT..." While the Inflorescence branching was indeed less affected by the HT in *qmib2^{MM}* in comparison to *qMIB2^{ST082}*, there is a clear increase in the inflorescence branches also in *qMIB2^{MM}* (Extended Data Fig. 2a). Therefore, I think the term less sensitive, rather the insensitive to HT will describe it better.

Response: Thank you for the suggestion. We have revised the description in this version.

10.In Fig. 2a, according to the primer position, the markers M4, M5, M6, M7 in the second row need to be changed to M5, M6, M7, M8. Please check.

Response: Thanks. We have revised the markers' names in this new version (Fig. 2a).

11. Extended Data Fig. 3c, please present the change in the protein sequence along with the difference in the DNA sequence.

Response: Thanks for your suggestion. We have marked the change in the protein sequence in the new version (Extended Data Fig. 3d).

12. Extended Data Fig. 3e, very low resolution; therefore, it was impossible to interpret the results.

Response: Thanks. We have changed this very low-resolution figure into a high-definition figure in the revised version (Extended Data Fig. 3g).

13. In the ChIP-seq experiment, the IgG control is missing in the SRA database. In addition, since MIB2-YFP-HA plants were used, a control using anti-HA on non-transgenic plants would be better as it accounts for the nonspecific binding of the HA antibody.

Response: We are sorry for this mistake. The IgG datasets have been deposited in the Sequence Read Archive (SRA) under the accession number SRR12756256. Indeed, using anti-HA on non-transgenic plants would be a good control. In the future, we would like prefer to use anti-HA in the control plants transformed with the empty vector.

14. Line 441, add that the Chip-seq data is also deposited.

Response: Thanks. We have revised it.

15. Line 174, please add the Solyc number for *SICOL1* and the same in the method section.

Response: Thanks. We have added the gene ID in the new version.

Reviewer #3 (Remarks to the Author):

The work reports genetic analysis of temperature-responsive inflorescence branching in tomato. The authors identified a major quantitative trait locus *MIB2* controlling inflorescence branching in response to high growth temperature. *MIB2* encodes a bHLH transcription factor orthologous to Arabidopsis SPATULA and its loss-of-function mutation as detected in MoneyMaker and some other cultivated tomato eliminates the potential of inflorescence branching when plants are grown under relatively high temperature. Using ChIP-seq and RNA-seq technology, the authors identified putative *MIB2* target genes and further showed that *MIB2* bound to the promoter region of *SICOL1*, a CONSTANS-like gene, to regulate its expression in response to high temperature. Based on the increased inflorescence branching in *SICOL1* overexpression in *mib2* -- NILs harboring MoneyMaker (MM) allele of the *MIB2* locus, the authors conclude that *MIB2*-*SICOL1* module play an important role in regulation of tomato inflorescence plasticity in response to temperature. It will be an interesting story after some key issues are solved, particularly those concerning the timing of *MIB2* and *SICOL1* action on inflorescence development in response to temperature.

Response: We appreciate the summary and comments for this research. Based on your suggestions, we have performed additional experiments and carefully revised the main text.

1. Solyc02g093280/bHLH017 was identified as *qMIB2* candidate; the MoneyMaker genome contains an insertion of the nucleotide G in its second exon disrupted coding potential, leading premature translation. Given the reference Heinz1706 genome has a complete *Solyc02g093280* and only small portion of BIG accessions contain this insertion, the MM allele of *qMIB2* was unlikely selected during tomato breeding. More data are needed to support the conclusion made in this study that *MIB2* was targeted by tomato breeders.

Response: We agree with your comments. Indeed, the G insertion is only found in about 30 big tomato accessions among 290 tomato accessions. We found that the tomato accessions taking MM allele of *qMIB2* have bigger fruits than other tomato accessions. Fruit size is always one of important traits for selection of the modern tomato variates. The *qMIB2* therefore might be selected during tomato breeding, although we have not more evidence to prove it. Thus, we revised our description into “the *qmib2*^{MM} locus may be selected during tomato breeding” in the new version.

2. The field experiments concerning inflorescence branching under normal and high temperature

(NT and HT) were conducted in two separate greenhouses located in two different places. Except temperature, other factors including soil, humidity and irrigation were varied between the two greenhouses. It is arguable that these factors or their interactions also had significant contribution to the observed variations in inflorescence branching. New experimental data collected from well-controlled growth conditions are required to demonstrate that temperature is the key factor determining this inflorescence plasticity.

Response: We gratefully appreciate your critical suggestions. As you said, the data we used in the manuscripts are collected in two separate greenhouses located in two different places. However, these two places are in the same latitude but at different altitude that ensure the plants are grown in the same day length, which is very important for floral transition. For guaranteeing the comparability of data, the management practices for fertilisation (including basing and top-dressing fertiliser) and irrigation are completely consistent during plant growth. At the same time, the corresponding control plants are grown in the same place with the mutants or the transgenic plants. Furthermore, we also compared the inflorescence branches of these plants in the same greenhouse under different ambient temperature and the results are below. All these results indicate that temperature is the key factor determining inflorescence plasticity.

3. The weedy *S. lycopersicum* var. *cerasiforme* (cherry tomato) is generally thought the admixture of domesticated tomato and its wild ancestor *S.pimpinellifolium*; its origin likely predates domestication (Ranc et al BMC Plant Biol 2008,8:130; Razifard et al Mol Biol Evol 2020,37:1118-1132). It is incorrect to call it "wild cherry tomato" (Line 38).

Response: Thanks. We have revised it to "cherry tomato".

4. The qMIB2 NILs were obtained by crossing RIL ST082 to MoneyMaker three times. Such NILs are not genetically pure enough for functional analysis of a particular trait locus, they still contained quite significant amount of genetic variations, affecting trait formation. Did the authors verify the genetic background of these NILs using DNA markers or genome survey by re-sequencing?

Response: Thanks for your comments. To ensure the identity of the NILs genome, we have tested the genetic polymorphisms between these two NILs using more than 100 DNA markers (8-10 for each chromosome). Our results showed that the genome similarity between the NILs was 98.02%. The PCR results are below and the related descriptions have been revised in the methods.

5. The inflorescence of NIL-*mib2*^{MM} showing in Figure 5h are highly branched, very different when comparing with the MM inflorescences showing in Figure 1a and 2b. Please clarify the discrepancy.

Response: Thank you for the comments. The different inflorescence branches between the NIL-*mib2*^{MM} and MM resulted from their different genetic backgrounds. The *STM3* and *J2* MADS-box proteins are two main factors in controlling inflorescence branching³⁻⁵. The *STM3*^D and *j2* alleles contribute to the increase of inflorescence branching. We tested the genotypes of these two genes and found that MM takes the *STM3*^D allele, and the NIL-*mib2*^{MM} carries the *j2* and *STM3*^D allele that leads to more inflorescence branches in the NIL-*mib2*^{MM} plants than that of MM plants.

6. Figure 3d-f presented ChIP-qPCR data to confirm MIB2 binding in *HSP90* and *HSFA6B* promoter regions. These data are not relevant to the question the authors want to address -- regulation of inflorescence development by MIB2, if the authors can't provide data to show the two genes are part of MIB2 regulatory network. These data may go to supplemental.

Response: Thank you for the suggestions. As a transcription factor responding to HT, the targets of MIB2 containing the heat shock protein genes are reasonable. *HSP90* and *HSFA6B* are directly bound by MIB2 and their transcription levels in the meristems are regulated by it, indicating that they are involved in MIB2 regulatory network. The results of *HSP90* and *HSFA6B* directly regulating by MIB2 confirmed its functions in high temperature response.

7. Under what growth conditions the two *SICOLI* overexpression lines showed inflorescence branching phenotype, NT or HT? Though *SICOLI* overexpression was sufficient to increase inflorescence branching, it is more important to know whether *SICOLI* is required for this trait. Inflorescence branching phenotypes of *scoli* mutants in *MIB2*^{CC} background will determine the

significance of *SICOL1* as MIB2 target in inflorescence branching.

Response: Thank you for the suggestion. Fig. 5h showed the inflorescence branches of the two *SICOL1* overexpression lines under HT conditions. We agree with your comments that mutation of *SICOL1* in the *MIB2^{CC}* background are the most direct evidence to confirm the function of *SICOL1*. Actually, we obtained the *scol1cr* mutants in the *NIL-MIB2^{ST082}* background.

Unfortunately, no significant inflorescence phenotypes were observed under NT and HT and the results are bellow (a). We then analyzed tomato genome and found that there are three homologs of Arabidopsis CO. One possibility is that these CO proteins are functional redundancy. To test this hypothesis, we obtained the double mutant of *scol1 scol2* in the *MIB2^{CC}* background and found that the number of inflorescence branches was significantly reduced compared with that of the wild type (b). These results indicated that *SICOLs* are required for the development of inflorescence branches as a direct target of MIB2.

8. The developmental stages of shoot meristems should be defined more specifically. The authors often just refer to meristems or reproductive meristematic tissues. It is unclear when MIB2 and *SICOL1* take into play during inflorescence development.

Response: Thanks. We agree with your opinion. The spatial and temporal expressions of genes are very important for their functions. According to previous results, the developmental stages of shoot meristem in tomato can be divided into five stages: VM (vegetable meristems); TM (transition meristems); FM: floral meristems); SIM (sympodial inflorescence meristems); SYM (sympodial meristems)⁶. To clarify MIB2 roles in inflorescence development, we detected the *MIB2* transcription in different meristematic tissues by *qPCR* and *in situ* hybridization. The results revealed that *MIB2* is expressed in various meristems and highly expressed in FMs (Extended Data Fig. 3b and c). We also tested the expression patterns of *SICOL1* and found that it is expressed in all meristem tissues and highly expressed in FMs and SIMs that is similar with MIB2 (Extended Data Fig. 5d). Thus, we used the “reproductive meristematic tissues” includes FMs and SIMs to perform our experiments.

References

- 1 Leivar, P. & Quail, P. H. PIFs: pivotal components in a cellular signaling hub. *Trends Plant Sci* **16**, 19-28, doi:10.1016/j.tplants.2010.08.003 (2011).
- 2 Toledo-Ortiz, G., Huq, E. & Quail, P. H. The Arabidopsis basic/helix-loop-helix transcription factor family. *Plant Cell* **15**, 1749-1770, doi:10.1105/tpc.013839 (2003).
- 3 Wang, X. *et al.* Antagonistic regulation of target genes by the SISTER OF TM3–JOINTLESS2 complex in tomato inflorescence branching. *The Plant Cell*, doi:10.1093/plcell/koad065 (2023).
- 4 Alonge, M. *et al.* Major Impacts of Widespread Structural Variation on Gene Expression and Crop Improvement in Tomato. *Cell* **182**, 145-+ (2020).
- 5 Soyk, S. *et al.* Bypassing Negative Epistasis on Yield in Tomato Imposed by a Domestication Gene. *Cell* **169**, 1142-1155 e1112, doi:10.1016/j.cell.2017.04.032 (2017).
- 6 Park, S. J., Jiang, K., Schatz, M. C. & Lippman, Z. B. Rate of meristem maturation determines inflorescence architecture in tomato. *Proc Natl Acad Sci U S A* **109**, 639-644, doi:10.1073/pnas.1114963109 (2012).

Reviewers' Comments:

Reviewer #1:

Remarks to the Author:

The authors have addressed the comments in my review to my satisfaction.

Reviewer #2:

Remarks to the Author:

All my comments were nicely addressed. Thank you.

I would suggest adding the expression of MIB2 in cotyledons and hypocotyl to the manuscript Figures or to the Sup. Figures (point 2 in the response to reviewer 2).

Reviewer #3:

Remarks to the Author:

All my concerns have been addressed.

Point-by-point responses to Reviewers' comments

Reviewer #1 (Remarks to the Author):

The authors have addressed the comments in my review to my satisfaction.

Response: Thank you so much for the encouragement.

Reviewer #2 (Remarks to the Author):

All my comments were nicely addressed. Thank you.

I would suggest adding the expression of MIB2 in cotyledons and hypocotyl to the manuscript Figures or to the Sup. Figures (point 2 in the response to reviewer 2).

Response: Thanks for your nice comments very much. We have added the expression of *MIB2* in cotyledons and hypocotyl in the Extended Data Fig. 5a according to your suggestion.

Reviewer #3 (Remarks to the Author):

All my concerns have been addressed.

Response: We sincerely appreciate your great support.